# In Vivo Toxicity Assessment of Chitosan-Coated Lignin Nanoparticles in Embryonic Zebrafish (*Danio rerio*)

**DOI:** 10.3390/nano11010111

**Published:** 2021-01-06

**Authors:** Jared S. Stine, Bryan J. Harper, Cathryn G. Conner, Orlin D. Velev, Stacey L. Harper

**Affiliations:** 1School of Chemical, Biological and Environmental Engineering, Oregon State University, Corvallis, OR 97331, USA; stinej@oregonstate.edu; 2Department of Environmental and Molecular Toxicology, Oregon State University, Corvallis, OR 97331, USA; bryan.harper@oregonstate.edu; 3Department of Chemical and Biomolecular Engineering, North Carolina State University, Raleigh, NC 27695, USA; cgconner@ncsu.edu (C.G.C.); odvelev@ncsu.edu (O.D.V.)

**Keywords:** toxicology, development, biocompatibility, nanotoxicology, sustainability

## Abstract

Lignin is the second most abundant biopolymer on Earth after cellulose. Since lignin breaks down in the environment naturally, lignin nanoparticles may serve as biodegradable carriers of biocidal actives with minimal environmental footprint compared to conventional antimicrobial formulations. Here, a lignin nanoparticle (LNP) coated with chitosan was engineered. Previous studies show both lignin and chitosan to exhibit antimicrobial properties. Another study showed that adding a chitosan coating can improve the adsorption of LNPs to biological samples by electrostatic adherence to oppositely charged surfaces. Our objective was to determine if these engineered particles would elicit toxicological responses, utilizing embryonic zebrafish toxicity assays. Zebrafish were exposed to nanoparticles with an intact chorionic membrane and with the chorion enzymatically removed to allow for direct contact of particles with the developing embryo. Both mortality and sublethal endpoints were analyzed. Mortality rates were significantly greater for chitosan-coated LNPs (Ch-LNPs) compared to plain LNPs and control groups. Significant sublethal endpoints were observed in groups exposed to Ch-LNPs with chorionic membranes intact. Our study indicated that engineered Ch-LNP formulations at high concentrations were more toxic than plain LNPs. Further study is warranted to fully understand the mechanisms of Ch-LNP toxicity.

## 1. Introduction

As the nanomaterials industry continues to grow and products containing nanomaterial-based biocides become more widely used, there is increasing concern over the environmental toxicity of nanomaterials [1]. Nanomaterials have proven to be difficult to remove from conventional water treatment facilities [2] and as metal-based antimicrobial nanomaterial applications become more widespread [3,4,5,6], potentially less toxic and less persistent material alternatives are needed to reduce the burden of persistent metal contaminants in the environment. One sustainable material currently being investigated is lignin, a biodegradable biopolymer found in trees that is widely available as a byproduct of the paper and biofuel industries [7,8]. A previous study looked at the efficacy of using biodegradable lignin nanoparticles (LNPs) as a vector to deliver biocidal actives for antimicrobial applications [9]. Other research indicates that lignin itself has its own antimicrobial properties. The presence of lignin extracts in coated limes was found to inhibit fungal growth [10]. The antibiotic effectiveness of Björkman beech lignin has been found to be comparable to some common antibiotics, including Bronopol and chlorhexidine [11]. Additionally, later research showed that the addition of LNPs to polyvinyl alcohol/chitosan hydrogels improved the overall effectiveness of the formulation against *Escherichia coli* [12]. These previous studies illustrate that nanomaterials containing lignin have the potential to be effective, less toxic, and biodegradable alternatives to conventional antimicrobial nanomaterials.

Developing sustainable agricultural products requires innovative solutions that utilize readily available, biocompatible, and compostable materials that can still perform as efficient treatments in protecting crops. Conventional fungicides are one class of products that often contain harsh chemicals that are environmentally toxic and expensive to produce. For example, copper sulfate is commonly used in organic agricultural applications to control fungal diseases among certain crops. As antifungal resistance has developed, copper nanoparticles have been posited to replace copper sulfate in these applications [6]. However, dissolved copper has been found to be very toxic at low concentrations in freshwater systems, particularly for salmonid species [13]. In order to reduce the metal burden on the environment and the development of antimicrobial resistance, more sustainable products need to be developed. Engineered LNPs could serve as the basis of a potential solution, and thus their environmental impact is evaluated here.

By applying principles of sustainable engineering during the design and synthesis of novel nanoparticle formulations, potentially undesirable environmental impacts may be avoided while still maintaining the desired nanoparticle characteristics [14]. For the engineered formulation in this study, LNPs were chosen for lignin’s natural biodegradability [15]. Previous research has also shown LNPs to be very biocompatible and have little or no impact on algae and yeast survival [16]. Additionally, it is possible that lignin can be precipitated into nanoparticles by using environmentally-benign solvents and the resulting nanoparticles are versatile enough to be used in a variety of applications.

Nanoparticles can be characterized according to their size, shape, chemical composition, surface structure, and surface charge [17,18,19,20]. External factors such as pH, temperature, ionic strength, and the presence of natural organic matter can impact the fate and transport of nanoparticles released in aquatic systems [21,22,23,24]. The interaction between the external factors and an individual nanoparticle’s characteristics can dictate, and potentially alter, nanoparticles’ properties such as bioavailability to living organisms [25,26]. To continue exploring the efficacy of LNPs as more sustainable alternatives to conventional antimicrobials, a new LNP formulation was engineered and evaluated here. A recent study investigated the use of chitosan nanoparticles to encapsulate conventional fungicides to more effectively combat basal stem rot in oil palm seedlings [27]. The new formulation described in this study uses LNPs with a chitosan coating (Ch-LNP) to enhance the adsorptive properties of particles to biological matter.

Chitosan is a naturally-occurring polysaccharide biopolymer derived from the deacetylation of chitin, the primary structural component in the outer skeleton of shellfish. Chitosan may have a varying fraction of C2 amino groups depending on the degree of deacetylation. The ability of these groups to become protonated and positively charged under weakly acidic conditions is thought to be the source of chitosan’s antimicrobial properties. The polycationic nature of chitosan favors interactions with negatively charged microbial cell walls [28], which can result in the disruption of cell membrane functions [29]. Other studies have reinforced this theory by showing chitosan to possess biocidal properties against *Streptococcus mutans* biofilms and *Propionibacterium acnes* [29,30]. In addition to these antimicrobial properties, chitosan’s surface charge, mucous adhesion, and anti-inflammatory properties have made it especially suitable for biomedical applications [29,31,32]. The engineered Ch-LNP formulation is expected to be a sustainable alternative to conventional antimicrobial products.

The goal of this study was to determine the biocompatibility or toxicity of plain LNPs and Ch-LNPs. We hypothesized that these particles would be low in toxicity to vertebrates based on previous literature showing that the individual components of the formulation, both LNPs and chitosan, are biocompatible [9,29,33]. To test this hypothesis, we utilized the embryonic zebrafish assay, a common model for assessing toxicity as it provides a variety of developmental endpoints that are critical to the survival of the organism [34,35]. Zebrafish develop quickly and are optically transparent, allowing for easy observations of phenotypic responses [34]. Zebrafish also share similar homology to humans, making observed effects of chemical stressors from this assay potentially indicative of human physiological responses [36]. The highly conserved nature of fundamental developmental processes in vertebrates across different species combined with their responsiveness to perturbation during the embryonic life stage makes the embryonic zebrafish a valuable model for evaluating the toxicity of nanomaterials in complex biological systems [37,38]. Additionally, the smaller assay volumes used in the embryonic zebrafish assay allow researchers to save on costs associated with generating large quantities of well-characterized nanomaterials, as compared to traditional rodent models. For these reasons, the embryonic zebrafish is a cost-effective and sensitive model for rapidly collecting data essential for understanding the biocompatibility of nanomaterials.

## 2. Materials and Methods

### 2.1. Materials and Characterization

The lignin used for the nanoparticle core is high purity lignin (HPL) provided by Lignol Innovation Corporation (Vancouver, BC, Canada). Stock nanoparticle suspensions were produced using antisolvent flash precipitation. The batch process begins with 1 mL of 0.05 g/mL lignin dissolved in acetone (a solvent). It was diluted very rapidly in 9.2 mL of deionized (DI) water that serves as an antisolvent and as a medium of the formed suspension [15]. The recirculation of the medium in a scaled up semi-continuous system allows high overall concentration of uniform nanoparticles to be achieved [39]. The Ch-LNPs were prepared through rapid mixing of a LNP dispersion with a chitosan solution. As chitosan is soluble under slightly acidic conditions, low molecular weight chitosan (Sigma-Aldrich, St. Luis, MO, USA) was dissolved in a solution of DI water with 1 vol% of acetic acid at a pH of approximately 3.5 to form the chitosan solution. During mixing with the more negatively charged LNP suspension, chitosan precipitates out of solution and the opposite charges (lignin particles being negative at −24 mV and chitosan being positive at +34 mV) lead to the adsorption of the chitosan on the LNPs. The LNPs and Ch-LNPs were maintained at a pH range of 3.8 to 4.1 and 5.0 to 5.1, respectively. The size distribution of the nanoparticle suspensions was determined using dynamic light scattering (DLS) with a Malvern Instruments Zetasizer Nano ZSP (Malvern, UK). The instrument has a 633 nm He-Ne laser and performed measurements with a 173° backscattering mode. The hydrodynamic diameter (HDD) for the LNPs and Ch-LNPs as synthesized were 75 ± 2 nm and 120 ± 5 nm, respectively.

Suspensions of LNPs and Ch-LNPs were dialyzed for 1 week to ensure that the trace amounts of the original solvent (acetone) were removed and thus the observed responses are a result solely of the particles present. Both LNPs and Ch-LNPs stock suspensions were dialyzed in a 10 kDa molecular weight cutoff membrane (SnakeSkin, Thermo Scientific, Waltham, MA, USA) placed in approximately 4 L of DI water for 7 days with gentle stirring and daily water changes to remove any remaining acetone or dissolved chitosan from the suspensions prior to performing dilutions for exposures. Both stock nanoparticle suspensions were stored in distilled water at 4 °C until use. Seven-fold dilutions of these stock nanoparticle suspensions were performed with simulated fishwater to prepare the various exposure concentration solutions. Fishwater was prepared by dissolving Instant Ocean salts (Aquatic Ecosystems, Apopka, FL, USA) in reverse osmosis water to a concentration of 260 mg/L and then adjusting the pH to 7.2 ± 0.2 and a measured conductivity value between 480 and 600 µS/cm using approximately 100 mg of sodium bicarbonate [35]. Fishwater ionic strength and hardness were measured at 0.02 mM and 36 mg/L, respectively. The engineered formulation of the Ch-LNPs were compared with plain LNPs to determine toxicological differences.

After being diluted with fishwater to 50 mg/L, HDD and average zeta potential for both nanoparticle suspensions were measured in triplicates using a Zetasizer Nano ZS (Malvern Instruments Ltd., Worcestershire, UK) at 26.8 °C. After these measurements were taken, 1 mL aliquots were stored in an incubator matching the conditions of the zebrafish embryos. Metadata associated with the zeta potential measurements can be found graphically depicted in Figure 1.

### 2.2. Embryonic Zebrafish Assay

The previously mentioned LNP and Ch-LNP nanoparticle suspensions were dispensed into two 96-well plates at different concentrations for each row of 12 wells. Exposure wells were filled with 200 µL of nanoparticle suspensions at increasing concentrations, while one row on each of the 96-well plates was a dedicated control containing only fishwater. The range of concentrations of each nanomaterial suspension used in this study were obtained using preliminary range-finding experiments to find the concentration at which 100% mortality was observed to determine the full concentration–response relationship for each nanoparticle suspension. Adult wild-type tropical 5D zebrafish (*Danio rerio*) were maintained in the Sinnhuber Aquatic Research Laboratory (SARL) at Oregon State University, Corvallis, OR, USA. Embryos received from the facility were between 6 and 8 h post-fertilization (hpf) and were inspected under a dissecting microscope prior to the study to ensure viability and developmental stage. The embryos were then individually placed into isolated wells in 96-well plates. Two groups were created for this study, one group with their chorionic membrane (chorion) preserved, while the other group of embryos had the chorion removed. Dechorionation was performed enzymatically by exposing groups of 200–400 embryos in glass petri dishes to 1.5 mL of a 50 mg/mL solution of protease from *Streptomyces griseus* for approximately 6 min until their chorions start to detach [40]. Since the chorion plays a role in protecting the embryo from external influences, removal of the chorion facilitates a toxicological assessment that is more indicative of direct environmental exposure [36]. Replicate plates were prepared and exposures were conducted simultaneously for each nanoparticle suspension, allowing us to have an overall sample size of 24 embryos per exposure concentration. Once samples were prepared, wells were covered with Parafilm to reduce evaporation and incubated at 26.8 °C under 14:10 light/dark cycles.

### 2.3. Embryonic Zebrafish Toxicological Evaluation

Embryos were analyzed at 24 hpf and 120 hpf for mortality and several developmental, morphological, and physiological abnormalities. Mortality, presence of spontaneous movement, delayed developmental progression, and notochord malformations were assessed at 24 hpf. Mortality, along with malformations of the snout, brain, pectoral and caudal fins, eyes, jaw, otic structures, axis, trunk, somites, swim bladder, and body pigmentation were assessed at 120 hpf. Physiological and behavioral endpoints were also assessed at 120 hpf including the presence of any pericardial or yolk-sac edema, impaired circulation, and active touch response [35]. Toxicological endpoints are reported in a binary format as either present or absent. Representative images of control embryos and individuals displaying abnormalities at 24 and 120 hpf were taken with an Olympus SZX10 microscope (Tokyo, Japan) equipped with an Olympus SC100 high resolution digital color camera (Olympus Corporation, Center Valley, PA, USA). Care was taken to ensure compliance with national care and use guidelines and approval of the Institutional Animal Care and Use Committee (IACUC) at Oregon State University (ACUP #5114).

### 2.4. Statistical Analysis

Statistical analyses were performed using SigmaPlot version 13.0 (Systat Software, San Jose, CA, USA) unless specified otherwise. Differences between exposed groups and controls were considered significant when *p* ≤ 0.05. Significant differences in HDD and zeta potential measurements were determined with repeated measures analysis of variance (ANOVA) and Tukey’s post hoc analysis. Two-way ANOVA was performed to ensure there were no significant differences in mortality between replicate exposure plates prior to combining the data. Mortality–response curves were generated as sigmoidal curves for the particles generating more than 50 percent mortality to determine their respective LC_50_ values. The lowest observable adverse effect levels (LOAEL) for all endpoints were determined using the first statistically significant deviation from the control groups as determined by comparing exposure groups with the control group using Fisher’s Exact Test (two-tailed). From the LOAEL, the highest concentration at which no observable adverse effects (NOAEL) were observed was determined by using the closest measured exposure concentrations below the noted LOAEL.

## 3. Results

### 3.1. Particle Characterization

DLS measurements of HDD and zeta potential were taken for the two nanoparticle suspensions in both DI water and after suspension in fishwater, as shown in Table 1. As expected, the zeta potential of the LNPs was negative, while the zeta potential of the Ch-LNPs was positive. The initial zeta potentials of the LNPs and Ch-LNPs in DI water were −20.1 ± 6.4 mV and 32.6 ± 4.9 mV, respectively. After the nanoparticle suspensions in fishwater (0.02 mM) were prepared, the zeta potentials of the LNPs and Ch-LNPs were measured at −9.9 ± 0.6 mV and 10.6 ± 0.3 mV, respectively. The HDD and zeta potential values of dialyzed nanoparticle suspensions in fishwater varied slightly for the plain LNPs and considerably for the Ch-LNPs when compared to the non-dialyzed suspensions in DI water. In general, Ch-LNPs were approximately double the size of the LNPs due to their chitosan coating. The initial measured HDD values for the LNPs and Ch-LNPs in DI water were 79 ± 3.6 nm and 129 ± 5.9 nm, respectively. The LNP and Ch-LNP solutions in fishwater had an HDD range of 91 ± 1.6 nm and 219 ± 7.6 nm, respectively. These differences in HDD are likely attributed to the small increase in salts present in the simulated fishwater which may lead to agglomeration through partial screening of the electrostatic double layer [1,15]. This is also shown by the zeta potential measurements for the nanoparticle suspensions in fishwater being closer to zero, indicating weaker electrostatic repulsion and making the particles more likely to agglomerate [41].

### 3.2. Formulation Toxicity Analysis

The toxicological exposures were performed using only dialyzed nanoparticles while comparing the impacts of LNPs and Ch-LNPs against dechorionated and chorion intact embryos. Once replicate exposure plates were found to have no significant differences in response (*p* ≤ 0.05), they were pooled to increase the sample size to 24 embryos per concentration, for each nanoparticle type tested. Observed and predicted concentration-response curves for the four study groups are shown in Figure 2. The LNP exposure groups, both in the presence and absence of the chorion, did not show any significant increase in 120 hpf mortality when compared to controls. Since there were no significant deviations in mortality from controls even at high concentrations, reported NOAELs for the plain LNPs would be greater than the highest exposure concentrations tested in this study. However, significant mortality was observed in the groups exposed to Ch-LNPs in both the presence (LC_50_ = 548 mg/L) and absence (LC_50_ = 105 mg/L) of the chorion when compared to controls (Table 2). Measured NOAEL concentrations for mortality in the Ch-LNP exposure group for the chorion-on scenario at 24 hpf and 120 hpf were of 1280 mg/L and 320 mg/L, respectively. We observed 100% mortality in the other Ch-LNP exposure group for the chorion-off scenario at 24 hpf. The measured NOAEL concentration for mortality in this exposure group was 40 mg/L. Based on these values it is evident that the Ch-LNPs were significantly more toxic than the LNPs at high concentrations and the toxicity of the Ch-LNPs was increased in the absence of the chorionic membrane. In both the presence and absence of the chorion, significant mortality at 24 hpf was observed in the Ch-LNP exposure groups. 

The presence of a chorion acting as a toxicological buffer is validated in other studies where removing the chorion was shown to increase toxic responses [36,42]. In one previous study, the chorion was shown to modulate silver toxicity by preventing the silver ions from entering the perivitelline fluid [42]. The presence of the chorion likely played a role in modulating the toxicity of the Ch-LNPs. Excessive hardness of the exposure media may contribute to differences in toxicity. The hardness of prepared fishwater may alter the characteristics of the Ch-LNPs, potentially impacting their overall toxicity. However, as the hardness of our fishwater was approximately 36 mg/L, significant changes in Ch-LNP toxicity as a result of water hardness are not likely. Typically, water with an equivalent calcium carbonate level of less than 60 mg/L can be categorized as soft water [43]. Many other studies have utilized moderately hard to hard water with equivalent calcium carbonate levels of up to 148 mg/L when exposing zebrafish. Some reports show that LC_50_ values can be higher in the presence of dissolved organic matter, followed by chlorine, sodium, and calcium ions [43]. This has been attributed to the coalescence effect, which can lead to complexation or the formation of nanoparticle agglomerates, which can decrease particle uptake by organisms reducing apparent nanoparticle toxicity [44].

Based on the data collected, Ch-LNPs are significantly more toxic than plain LNPs. There was a 2:1 ratio of lignin to chitosan in the Ch-LNP formulation by weight, meaning the mass of the chitosan coating in each particle was approximately one third of the overall mass of each individual nanoparticle. The Cryo-SEM image in Figure 3, shows that Ch-LNPs are generally uniform and spherical. This suggests that the LNP cores are uniformly coated with chitosan layers, and the toxicity values in the Ch-LNPs are likely heavily influenced by the chitosan coating. The major change in the particle properties induced by the chitosan adsorption is the “re-charging” of their surface potential from the original negative value, to a positive one. This could make them adhere strongly to live cells and biological tissues, which commonly have negative surface charges at normal pH conditions. The strong adherence of positively charged nanoparticles to negatively charged cell membranes is likely leading to the disruption of transmembrane transport and mechanical straining, which could affect profoundly the metabolic activity and viability of single cellular organism [2,15,45]. The impacts on more complex organisms such as zebrafish embryos have yet to be understood in detail.

### 3.3. Sublethal Endpoints Analysis

Among the sublethal toxicological endpoints evaluated, only yolk-sac edema was found to be significant (*p* ≤ 0.05). Based on our experimental data, the plain LNPs did not elicit any significant sublethal impacts. The rapid mortality seen after exposure to the Ch-LNPs likely prevented a number of significant sublethal impacts from being observed. For the group exposed to Ch-LNPs at 320 mg/L in the chorion-on scenario, significant increases in the incidence of yolk-sac edema were observed just before significant mortality occurred at the next highest exposure concentration (Figure 4). Representative images shown in Figure 4b show embryonic zebrafish with developmental abnormalities across the concentration range at which sublethal toxicological endpoints could be observed in the group exposed to Ch-LNPs with chorion intact. Yolk-sac edemas for this study group had a measured NOAEL of 160 mg/L at 120 hpf. Yolk-sac edema could be indicative of oxygen deprivation in the embryo due to particles clogging or otherwise restricting gas exchange through the pores in the chorionic membrane [46]. Decreases in the occurrence of yolk-sac edema for the Ch-LNP exposure group in the presence of the chorion at higher concentrations are likely an artifact of the increased mortality seen at higher concentrations, reducing the number of viable embryos for sublethal endpoint analysis (Figure 4c,d). Although not statistically significant, it should also be noted that at this concentration there were increased observations of malformations, including snout, jaw, heart, and pectoral fin.

The circulatory system and heart in zebrafish embryos typically begin to form between 21 and 24 hpf [47,48]. In the embryos that survived beyond 24 hpf in the chorion-on scenario, increasing exposure concentrations to Ch-LNPs increased the likelihood of observed sublethal impacts. As previously mentioned, other studies have shown that the chorion plays a role in modulating toxicity [42]. There is also evidence of nanoparticles between 30 and 72 nm passing through the chorionic membrane and being distributed to various parts of the fish, including the brain, heart, yolk, and blood [49]. Distribution of particles to the yolk and heart could lead to interference with osmoregulation and cause edemas [50,51,52,53]. Disturbing these developmental pathways during early development may disrupt embryo growth, resulting in malformations [54]. Agglomeration of nanoparticles in exposure media could potentially alter the oxygen exchange through chorionic pores and impact the osmotic balance in the embryo, leading to edemas similar to those observed [49].

Similar to previous studies using functionalized LNPs, neither the plain LNP nor Ch-LNP exposures induced delayed hatching. Significant delays in hatching and developmental progression, if observed, could have been attributed to high concentrations of plain LNPs [9]. Developmental progression has been found to be impaired following exposures of high concentrations (350 mg/L or higher) of plain LNPs due to their interactions with ions necessary for embryo development [9]. This is further supported by the finding that this process does not occur when LNPs are functionalized with a coating, which would occupy available ionic binding sites on the particles [9].

## 4. Discussion

DLS measurements were used to characterize particle size as electron microscopy techniques require drying of the nanoparticle suspensions which may contribute to agglomeration of the particles. Based on the measurements conducted, the LNP and Ch-LNP suspensions in fishwater media may be considered stable. Zeta potential measurements provide an indication of colloidal stability. Generally, a measured zeta potential magnitude greater than 30 mV indicates higher stability and magnitudes smaller than 5 mV are considered more likely to agglomerate [55]. Considering the magnitude of the zeta potential measurements for both nanoparticle suspensions in this study were approximately 10 mV, they likely would not have agglomerated significantly and would have remained uniformly distributed in the fishwater media within the exposure wells throughout the entire duration of study. Additionally, no visible particle sedimentation or agglomeration was observed.

Results from this study yield key insights into nanoparticle formulations engineered for sustainability. The data generated show that the inclusion of a chitosan coating in engineered LNPs posed increased toxicological risks to our sample population, but only at very high concentrations. Since these Ch-LNPs have not been previously studied for their toxicity, pure chitosan nanoparticles may provide a reasonable comparison. Our estimated Ch-LNP LC_50_ values, 105 mg/L in the absence of the chorion and 548 mg/L in the presence of the chorion, are roughly comparable to those observed under similar testing conditions using pure chitosan nanoparticles at similar sizes and concentrations. One study using embryonic zebrafish with their chorionic membranes intact reported an LC_50_ values of 270 mg/L at 120 hpf for pure chitosan nanoparticles 85 nm in diameter [56]. Another study exposed embryonic zebrafish with chorions removed to pure chitosan nanoparticles between 150 and 200 nm in diameter and observed no mortality below 200 mg/L [57]. The findings of these two studies differ with those of a study that reported significant mortality and a decrease in hatching rates in embryonic zebrafish with their chorion intact exposed to 200 nm pure chitosan nanoparticles at concentrations of 40 mg/L [58]. Other studies have found chitosan to be biocompatible and to have minimal toxicity [29,30,33]. Thus, it appears that the biological impact of chitosan in nanoparticle form may be strongly affected by its charge, molecular weight, and nanoscale morphology. These factors are still poorly characterized and merit further study.

Noting the increased toxicity of the Ch-LNPs and the fact that chitosan is a commonly used biopolymer that is typically considered non-toxic, further study of these particles is warranted. Utilizing similar experimental conditions, information related to pure chitosan nanoparticles could potentially be generated to help elucidate the mechanisms of toxicity of Ch-LNPs. Additional measurements related to oxidative stress, liver activity, and neurological function may further explain the degree to which Ch-LNPs may be toxic and to what extent. This additional data would further benefit the development of biodegradable nanoparticles, expand our knowledge of the benefits and limitations of engineered LNPs, and inform the design of other sustainable alternatives to existing engineered inorganic and metallic nanoparticles. As the concentrations tested in this study are likely unrealistically high, further testing of Ch-LNPs under environmentally-relevant conditions would provide additional insight into their mechanisms of toxicity.

## Figures and Tables

**Figure 1 nanomaterials-11-00111-f001:**
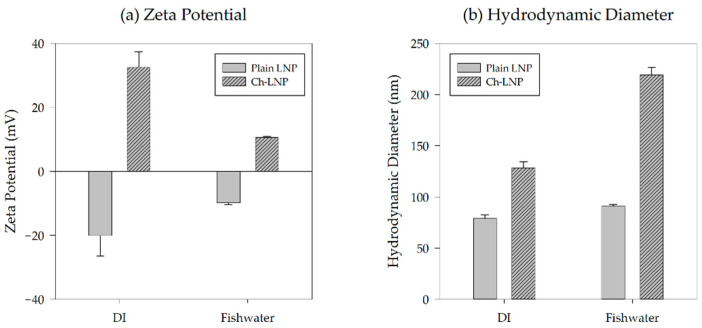
Analytical results showing (**a**) zeta potential and (**b**) HDD for the different particle formulations immediately after synthesis and after dispersal into fishwater prior to exposures.

**Figure 2 nanomaterials-11-00111-f002:**
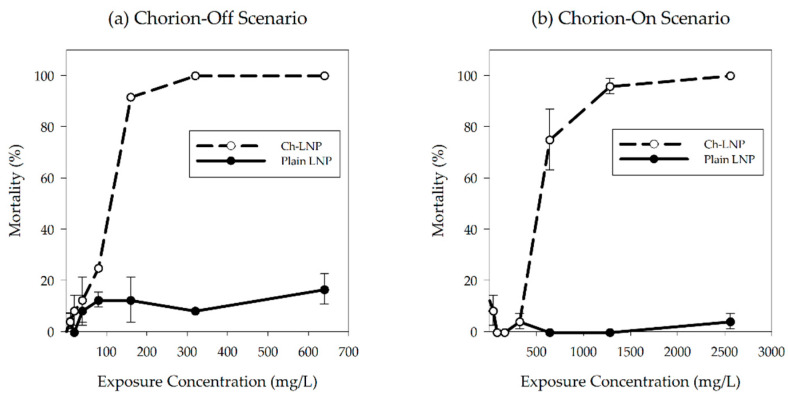
Percentage of observed mortality at 120 hpf for embryonic zebrafish exposed to particle suspensions with their (**a**) chorion removed and (**b**) with chorion intact; concentration-response curves showing predicted mortality at 120 hpf for the groups exposed to Ch-LNPs with (**c**) chorion removed and (**d**) chorion intact; a sigmoidal relationship was fit to experimental data to estimate LC_50_ values and 95 percent confidence intervals.

**Figure 3 nanomaterials-11-00111-f003:**
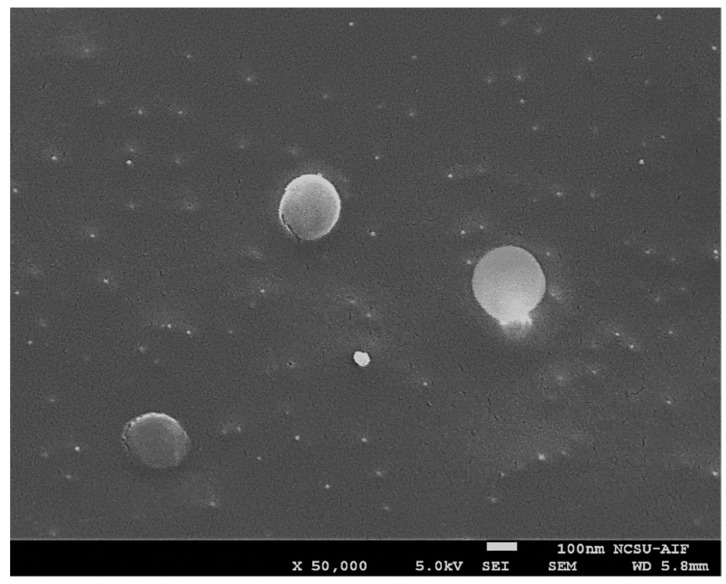
A Cryo-SEM image of the Ch-LNPs showing their approximate size and spherical shape.

**Figure 4 nanomaterials-11-00111-f004:**
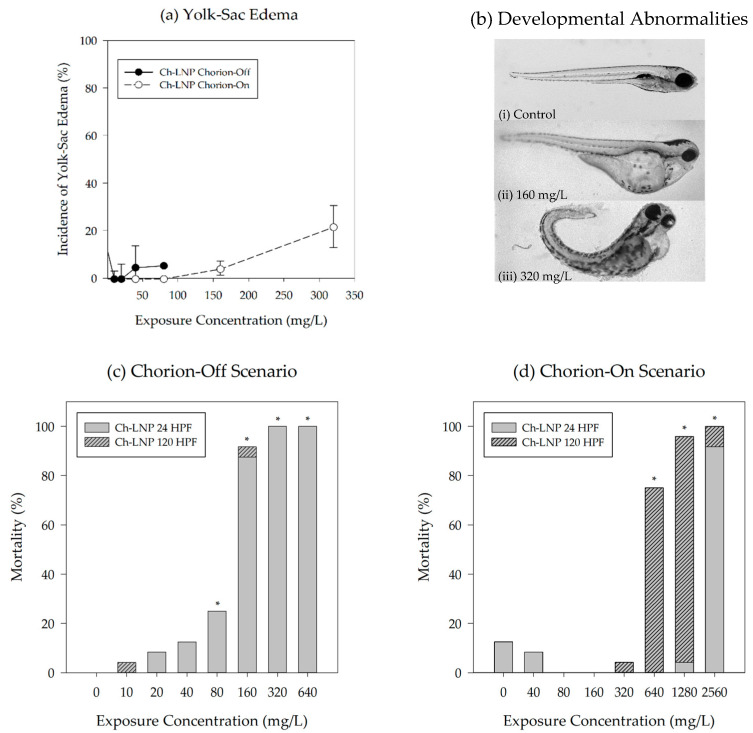
Incidence of yolk-sac abnormality observed at 120 hpf for (**a**) groups exposed to Ch-LNP suspensions showing data for both the chorion-on (dashed line) and chorion-off (solid line) scenarios; (**b**) representative images depicting developmental abnormalities in embryonic zebrafish 120 hpf from (**i**) 0 mg/L, (**ii**) 160 mg/L, and (**iii**) 320 mg/L Ch-LNP exposures with chorion intact; data for the chorion-off scenario are not available past concentrations of 100 mg/L due to high mortality seen at 24 hpf as shown by the observed mortality for both particle formulations during exposures with (**c**) chorion removed and (**d**) chorion intact; observations at both 24 hpf and 120 hpf shown together and an asterisk (*) represents a significant increase in mortality relative to control at *p* ≤ 0.05.

**Table 1 nanomaterials-11-00111-t001:** Dynamic light scattering (DLS) results showing zeta potential and hydrodynamic diameter (HDD) for the different nanoparticle suspensions.

Particle	Solution	HDD (nm)	Zeta Potential (mV)	Polydispersity Index
LNP	DI water	79 ± 3.6	−20.1 ± 6.4	0.165 ± 0.006
	Fishwater	91 ± 1.6	−9.9 ± 0.6	0.333 ± 0.023
Ch-LNP	DI water	129 ± 5.9	32.6 ± 4.9	0.272 ± 0.007
	Fishwater	219 ± 7.6	10.6 ± 0.3	0.393 ± 0.006

**Table 2 nanomaterials-11-00111-t002:** Calculated LC_50_ values at 120 hpf for Ch-LNPs in both experimental scenarios.

Particle	Scenario	LC_50_ (mg/L)	*R* ^2^
Ch-LNP	Chorion-Off	105	0.996
	Chorion-On	548	0.984

## Data Availability

The data presented in this study are available in the Nanomaterial-Biological Interactions knowledgebase (http://nbi.oregonstate.edu/).

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
