# Peer review of "In Vivo Toxicity Assessment of Chitosan-Coated Lignin Nanoparticles in Embryonic Zebrafish (Danio rerio)"

_nanomaterials, 2021, doi:10.3390/nano11010111_

Round 1

Reviewer 1 Report

The Authors have submitted a manuscript regarding the toxicity evaluation of chitosan coated lignin nanoparticles in zebrafish.

Investigations on nanotoxicology are pivotal for the advancement of nanomedicine as well as for risks assessment. Thus, the research addressed in this manuscript is of interest. The claims are supported and the exposition enough professional. This manuscript seems to meet the standards required by the Journal Policy. Overall, I suggest its acceptance for publication after the following minors are addressed:

-Authors should better discuss why they have employed that range of concentration of nanoparticles.

-Authors should better discuss the importance of zebrafish in the preclinical research (see for example, doi: 10.1016/j.jconrel.2019.08.022, 10.1021/acsami.0c17492)

-In order to complete the investigation and to make more appealing this manuscript, the Authors should supply other analysis of zebrafish such as hatching rate, malformations, heart rate and distance of swimming (see for example, doi: 10.1080/17435390.2018.1498551) together with the survival rate.

-Fig. 1 and 2: the error bars are missed.

- From my point of view, some SI images should be reported in the main text (for example the images of zebrafish) in order to compose a more informative manuscript.

Author Response

Response to Reviewer 1

Point 1: Authors should better discuss why they have employed that range of concentration of nanoparticles.

Response 1: Statement added in Section 2.2 to say, “The range of concentrations of each nanomaterial suspension used in this study were obtained using preliminary range-finding experiments to find the concentration at which 100 percent mortality was observed to determine the full concentration-response relationship for each nanoparticle suspension.”

Point 2: Authors should better discuss the importance of zebrafish in the preclinical research (see for example, doi: 10.1016/j.jconrel.2019.08.022, 10.1021/acsami.0c17492)

Response 2: More detail added to Introduction to elaborate on the value of the zebrafish model. Language added states, “The highly conserved nature of fundamental developmental processes in vertebrates across different species combined with their responsiveness to perturbation during the embryonic life stage makes the embryonic zebrafish a valuable model for evaluating the toxicity of nanomaterials in complex biological systems [37,38]. Additionally, the smaller assay volumes used in the embryonic zebrafish assay allow researchers to save on costs associated with generating large quantities of well-characterized nanomaterials, as compared to traditional rodent models. For these reasons, the embryonic zebrafish is a cost-effective and sensitive model for rapidly collecting data essential for understanding the biocompatibility of nanomaterials.”

Point 3: In order to complete the investigation and to make more appealing this manuscript, the Authors should supply other analysis of zebrafish such as hatching rate, malformations, heart rate and distance of swimming (see for example, doi: 10.1080/17435390.2018.1498551) together with the survival rate.

Response 3: Lines 151-158 of the originally submitted manuscript state that we evaluated malformations and developmental endpoints. We did not assess heart rate or distance of swimming. Revised the first sentence of Section 3.3 for clarity stating, “Among the sublethal toxicological endpoints evaluated, only yolk-sac edema was found to be significant (p ≤ 0.05).” There is also a sentence at the end of Section 3.3 stating, “Although not statistically significant, it should also be noted that at this concentration there were increased observations of malformations, including snout, jaw, heart, and pectoral fin.”

Point 4: Fig. 1 and 2: the error bars are missed.

Response 4: Revised figures to include error bars.

Point 5: From my point of view, some SI images should be reported in the main text (for example the images of zebrafish) in order to compose a more informative manuscript.

Response 5: Incorporated a new table with DLS data and moved the zebrafish images and cryo-SEM images into the main body of the manuscript.

Reviewer 2 Report

The matter of the manuscript is in my opinion interesting and important. Provided data and message is straightforward and simple. The introduction is very comprehensive but lack of depth of discussion is big fault.

Despite of overall good reception I have few important comments and issues that need to be clarified and improved.

Major:

The Methods or Results sections should state:

1. Are both nanoparticles similar in size?

2. Are both nanoparticles of similar density?

3. Are both nanoparticles sediment or not sediment in same way?

4. How many embryos per well were used?

5. How many embryos were used in single group in the experiment?

6. How may experiments were performed?

7. The EC50 values should be provided in separate table along with error measurement (e.g R2 of fitted curve?).

The toxicity curve is sigmoidal and small variations to one or two points can greatly affect the EC50 value.

8. Dose need to clarified. Are the nanoparticles sediment on the bottom? If yes the dose should be expressed in weight units per area. It is the better indication of dose as the interaction occurred on the bottom of the well.

9. The results need to be discussed better.

How the experimental set up may affect the toxicity?

What are the proposed or mechanisms of toxicity?

To what extent the amount of nanomaterials studied reflects the environmental conditions?

Can impurities form NPs synthesis affects the viability of embryos?

Is it possible that in high concentrations the nanoparticles affects the gas exchange?

Author Response

Response to Reviewer 2

Point 1: Lack of depth of discussion is big fault.

Response 1: Added another paragraph to the beginning of the discussion section to include a particle characterization analysis. New paragraph states:

“DLS measurements were used to characterize particle size as electron microscopy techniques require drying of the nanoparticle suspensions which may contribute to agglomeration of the particles. Based on the measurements conducted, the LNP and Ch-LNP suspensions in fishwater media may be considered stable. Zeta potential measurements provide an indication of colloidal stability. Generally, a measured zeta potential magnitude greater than 30 mV indicates higher stability and magnitudes smaller than 5 mV are considered more likely to agglomerate [55]. Considering the magnitude of the zeta potential measurements for both nanoparticle suspensions in this study were approximately 10 mV, they likely would not have agglomerated significantly and would have remained uniformly distributed in the fishwater media within the exposure wells throughout the entire duration of study. Additionally, no visible particle sedimentation or agglomeration was observed.”

Point 2: Are both nanoparticles similar in size?

Response 2: Lines 189-191 of the originally submitted manuscript state the sizes of the particles. I have included a table summarizing this data as well as a clarifying statement in Section 3.1 saying, “In general, Ch-LNPs were approximately double the size of the LNPs due to their chitosan coating.”

Point 3: Are both nanoparticles of similar density?

Response 3: Lines 235-238 of the originally submitted manuscript state that the Ch-LNPs contain a 2:1 lignin to chitosan weight ratio.

Point 4: Are both nanoparticles sediment or not sediment in same way?

Response 4: Language was included in the paragraph added to the beginning of the Discussion section to address this comment. Generally, both types of nanoparticles were considered stable suspensions.

Point 5: How many embryos per well were used?

Response 5: One embryo per well was used. Line 140 of the originally submitted manuscript states that the “embryos were individually placed into isolated wells”.

Point 6: How many embryos were used in single group in the experiment?

Response 6: Lines 147-149 of the originally submitted manuscript state that “replicate plates were prepared and exposures were conducted simultaneously for each nanoparticle suspension, allowing us to have an overall sample size of 24 fish per exposure concentration”. I have revised the term “fish” to state “embryos” throughout the manuscript for clarification.

Point 7: How may experiments were performed?

Response 7: Lines 147-148 of the originally submitted manuscript state that “replicate plates were prepared and exposures were conducted simultaneously for each nanoparticle suspension”. A total of four 96-well plates were prepared for embryonic zebrafish assays with 8 different concentrations used (including controls). One experiment assessed the toxicity of LNPs with two plates and the other experiment assessed the toxicity of Ch-LNPs with two plates.

Point 8: The EC50 values should be provided in separate table along with error measurement (e.g R2 of fitted curve?). The toxicity curve is sigmoidal and small variations to one or two points can greatly affect the EC50 value.

Response 8: A table displaying calculated LC50 values has been added to Section 3.2 that includes R2 values for clarity. Additionally, a 95 percent confidence interval band is displayed in Figure S3 in the Supplemental Information.

Point 9: Dose need to clarified. Are the nanoparticles sediment on the bottom? If yes the dose should be expressed in weight units per area. It is the better indication of dose as the interaction occurred on the bottom of the well.

Response 9: Language was included in the paragraph added to the beginning of the Discussion section to address this comment. The nanoparticles are considered uniformly distributed within the exposure media.

Point 10: The results need to be discussed better.

Response 10: Made some minor revisions throughout the Results section for clarity. Additionally, a paragraph elaborating on the particle exposures was added to the beginning of the Discussions section to address this comment.

Point 11: How the experimental set up may affect the toxicity?

Response 11: The originally submitted manuscript discusses the experimental set up including how dialyzing particles removes excess material that might contribute to toxicity (Lines 112-118). Difference between chorion-on and chorion-off toxicity is also discussed (Lines 145-147). The role of the chorion is further elaborated in Lines 221-225. I have revised the discussion relating to hardness in Lines 225-228 for clarity to state, “Excessive hardness of the exposure media may contribute to differences in toxicity. The hardness of prepared fishwater may alter the characteristics of the Ch-LNPs, potentially impacting their overall toxicity. However, as the hardness of our fishwater was approximately 36 mg/L, significant changes in Ch-LNP toxicity as a result of water hardness are not likely.”

Point 12: What are the proposed or mechanisms of toxicity?

Response 12: Some proposed mechanisms of toxicity for Ch-LNPs are noted in Lines 239-247 in the originally submitted manuscript. Additionally, some mechanisms of toxicity for developmental abnormalities in the presence of a chorion are noted in Lines 252-260 of the original submission.

Point 13: To what extent the amount of nanomaterials studied reflects the environmental conditions?

Response 13: The concentrations of nanomaterials studied were used to assess the extent to which the materials are toxic. Lines 315-317 in the originally submitted manuscript imply that the conditions tested in this study reached significantly high concentrations that are not environmentally relevant. I have added some language to the last sentence of the Discussion for clarity by stating, “As the concentrations tested in this study are likely unrealistically high, further testing of Ch-LNPs under environmentally-relevant conditions would provide additional insight into their mechanisms of toxicity.”

Point 14: Can impurities form NPs synthesis affects the viability of embryos?

Response 14: The synthesis with flash nanoprecipitation is more physical rather than chemical. The only impurities that could possibly form in the process are irregularly sized particles or aggregates, both of which settle out of suspension over time, making them a non-issue. Although they likely would not be removed by dialysis, they likely would not make it to the dialysis tubing as they would have settled to the bottom of the suspension. The second paragraph in Section 2.1 discusses how the nanoparticle formulations are dialyzed to remove any excess acetone such that the responses are solely a result of the particles themselves.

Point 15: Is it possible that in high concentrations the nanoparticles affects the gas exchange?

Response 15: Section 3.3 discusses how high concentrations of nanoparticles may affect gas exchange by stating, “Yolk-sac edema could be indicative of oxygen deprivation in the embryo due to particles clogging or otherwise restricting gas exchange through the pores in the chorionic membrane [46].” Language from the originally submitted manuscript state “aqueous exchange” and this was revised to “gas exchange” for clarity.

Reviewer 3 Report

In this contribution by Stine and co-workers, the authors investigated in vivo toxicity assessment of chitosan-coated lignin nanoparticles in embryonic zebrafish. In particular, I suspect that lots of information are missing, and it will require huge amount of additional work before it can be recommended for publication in Nanomaterials.

1) The discussion is too short and there is no conclusion in this ms.

2) From my experience, in the solution of ionic strength of 0.02 mM (or even less), the zeta potential of the chitosan-coated nanoparticles (the zeta potentials in water reported in the ms. was around 30 mV) should not be closed to zero.

3) One of the most important factors for a scientific article is that it should be reproducible by others. Although many experimental details have been provided, certain information should also be included. For instance, how is the chitosan-lignin nanoparticles prepared? What is the concentration of chitosan? How is the chitosan prepared? What’s the details of zebrafish maintenance?

4) A table of the summary about the size, PDI and zeta potential before and after modification (or in different media) should be added.

5) DLS result should be shown as a figure, not only mentioning the numbers.

6) TEM result would be very helpful to see whether chitosan-coated nanoparticles is aggregate as the authors thought or not.

Author Response

Response to Reviewer 3

Point 1: The discussion is too short and there is no conclusion in this manuscript.

Response 1: The guidelines in the Nanomaterials manuscript template do not require a Conclusion section and state, “This section is not mandatory, but can be added to the manuscript if the discussion is unusually long or complex.” Since the discussion is not unusually long or complex, some brief concluding statements were included in the discussion section. Additionally, I have added more detail to the beginning of the Discussion section to provide more depth on particle characterization. The new paragraph states:

 “DLS measurements were used to characterize particle size as electron microscopy techniques require drying of the nanoparticle suspensions which may contribute to agglomeration of the particles. Based on the measurements conducted, the LNP and Ch-LNP suspensions in fishwater media may be considered stable. Zeta potential measurements provide an indication of colloidal stability. Generally, a measured zeta potential magnitude greater than 30 mV indicates higher stability and magnitudes smaller than 5 mV are considered more likely to agglomerate [55]. Considering the magnitude of the zeta potential measurements for both nanoparticle suspensions in this study were approximately 10 mV, they likely would not have agglomerated significantly and would have remained uniformly distributed in the fishwater media within the exposure wells throughout the entire duration of study. Additionally, no visible particle sedimentation or agglomeration was observed.”

Point 2: From my experience, in the solution of ionic strength of 0.02 mM (or even less), the zeta potential of the chitosan-coated nanoparticles (the zeta potentials in water reported in the manuscript was around 30 mV) should not be closed to zero.

Response 2: In order to compare the chitosan-coated lignin nanoparticles to other chitosan-coated nanoparticles, I would need additional information pertaining to the nanoparticle suspension being compared. There could be other factors contributing to the difference in zeta potential measurements. Some information regarding nanoparticle size, concentration, synthesis methods, pH of solution, and media composition might be needed. Additionally, the nanoparticles in our study were also dialyzed for a week which may have also contributed to differences in measured zeta potential. Figure 8 in Richter et al., 2016 cited in the original manuscript shows how similarly synthesized tunable LNPs are affected by changes in ionic strength. Another citation was added for this Richter et al., 2016 in Section 3.1 of the manuscript for clarity.

Point 3: One of the most important factors for a scientific article is that it should be reproducible by others. Although many experimental details have been provided, certain information should also be included. For instance, how is the chitosan-lignin nanoparticles prepared? What is the concentration of chitosan? How is the chitosan prepared? What’s the details of zebrafish maintenance?

Response 3: An in-depth discussion of tunable LNP synthesis is detailed in Richter et al., 2016 cited in Section 2.1 of the originally submitted manuscript. Details regarding the chitosan solution mixed with the LNP suspension have been added to Section 2.1 to enhance the reproducibility of our work.

Lines 136-140 of the originally submitted manuscript state that we received zebrafish embryos from a separate facility between 6 and 8 hours post-fertilization. Zebrafish embryos up to 120 hours post-fertilization subsist on nutrients from their yolk-sac and therefore do not require feeding. Additionally, Lines 140-145 in the original submission state how embryos were dechorionated. Lines 149-150 explain how embryos are maintained by covering wells in parafilm to prevent evaporation and incubated at 26.8°C under 14:10 light/dark cycles.

Point 4: A table of the summary about the size, PDI and zeta potential before and after modification (or in different media) should be added.

Response 4: A table summarizing HDD, zeta potential, and PDI data for both nanoparticle formulations in different media has been added to Section 3.1 for clarity. Additionally, Figure S1 in the Supplemental Information shows HDD and zeta potential data graphically.

Point 5: DLS result should be shown as a figure, not only mentioning the numbers.

Response 5: A table summarizing HDD, zeta potential, and PDI data for both nanoparticle formulations in different media has been added to Section 3.1 for clarity. Additionally, Figure S1 in the Supplemental Information shows HDD and zeta potential data graphically.

Point 6: TEM result would be very helpful to see whether chitosan-coated nanoparticles is aggregate as the authors thought or not.

Response 6: A representative cryo-SEM image of the Ch-LNPs was pulled from the Supplemental Information and added to the main body of the manuscript for clarity. Language regarding our use of DLS instead of TEM imaging for particle characterization was added to the first paragraph of the Discussion section for additional clarity.

Round 2

Reviewer 3 Report

The manuscript entitled 'In Vivo Toxicity Assessment of Chitosan-Coated Lignin Nanoparticles in Embryonic Zebrafish (Danio rerio)' can be accepted for publication in the present form now.

Author Response

Response to Reviewer 3

Point 7: The manuscript entitled 'In Vivo Toxicity Assessment of Chitosan-Coated Lignin Nanoparticles in Embryonic Zebrafish (Danio rerio)' can be accepted for publication in the present form now.

Response 7: Additional changes to the figures in the main manuscript have been made to further enhance the readability of the overall manuscript.